# L-Carnitine and Chronic Kidney Disease: A Comprehensive Review on Nutrition and Health Perspectives

**DOI:** 10.3390/jpm13020298

**Published:** 2023-02-08

**Authors:** Bharti Sharma, Dinesh Kumar Yadav

**Affiliations:** 1Department of Pharmaceutical Sciences, College of Pharmacy and Health Sciences, St. John’s University, New York, NY 11439, USA; 2Department of Pharmacognosy, SGT College of Pharmacy, SGT University, Gurugram 122505, India

**Keywords:** Chronic Kidney Disease, L-carnitine, biosynthesis, biodegradation, health supplements, oxidative stress, inflammatory stress

## Abstract

Progressive segmental glomerulosclerosis is acknowledged as a characteristic of Chronic Kidney Disease (CKD). It is a major health issue that exponentially reduces health and economy and also causes serious morbidity and mortality across the globe. This review is aimed at comprehending the health perspectives of L-Carnitine (LC) as an adjuvant regimen for alleviating CKD and its associated complications. The data were gathered from different online databases such as Science Direct, Google Scholar, ACS publication, PubMed, Springer, etc., using keywords such as CKD/Kidney disease, current epidemiology and its prevalence, LC supplementations, sources of LC, anti-oxidant and anti-inflammatory potential of LC and its supplementation for mimicking the CKD and its associated problem, etc. Various items of literature concerning CKD were gathered and screened by experts based on their inclusion and exclusion criteria. The findings suggest that, among the different comorbidities such as oxidative stress and inflammatory stress, erythropoietin-resistant anemia, intradialytic hypotension, muscle weakness, myalgia, etc., are considered as the most significant onset symptoms in CKD or hemodialysis patients. LC or creatine supplementation provides an effective adjuvant or therapeutic regimen that significantly reduces oxidative and inflammatory stress and erythropoietin-resistant anemia and evades comorbidities such as tiredness, impaired cognition, muscle weakness, myalgia, and muscle wasting. However, no significant changes were found in biochemical alteration such as creatinine, uric acid, urea, etc., after creatine supplementation in a patient with renal dysfunction. The expert-recommended dose of LC or creatine to a patient is approached for better outcomes of LC as a nutritional therapy regimen for CKD-associated complications. Hence, it can be suggested that LC provides an effective nutritional therapy to ameliorate impaired biochemicals and kidney function and to treat CKD and its associated complications.

## 1. Introduction

Progressive segmental glomerulosclerosis or extreme loss of glomerulus filtration rate are generally acknowledged as the main characteristics of CKD [1]. Environmental toxins, drug toxins, or biologically induced toxins affect kidney function drastically; the reported common pathophysiology of kidney disease concerns the excessive production of endogenous oxidants such as reactive nitrogen species (RNS) and reactive oxygen species (ROS) as well as inflammatory cytokines. However, in most cases, nephrotoxicants substantially trigger such endogenous agents, which suppress kidney function [2]. Moreover, the burden of CKD, its high extent, and its influences are better defined in different developed countries. However, growing concern regarding CKD is progressively encroaching on the discretion of the healthcare system and researchers to curb kidney disease or associated disorders. As per the report published by the Center for Disease Control and Prevention (CDCP) on CKD in the United States, in 2021, more than 1 person out of 7, or about 15% of adults, 37 million people, are projected to have CKD. Furthermore, it is also reported that 9 out of 10 adults are associated with CKD, though they are not aware that they have CKD. Conclusively, it has been considered that out of 5 adults, 2 people are associated with progressive segmental glomerulitis [3]. An exponential growth of CKD has been seen in Asia, with approximately >4.5 billion people, or 60% of the total population globally. Considering these facts regarding South Asia alone, most diabetes prevalence is associated with the increasing risk of CKD. In China and India, the projection is estimated to remain as these countries have the highest number of people. Moreover, increasing clusters of CKD and associated disorders with unknown aetiology have been reported in parts of Asia. However, the data on the contemporary prevalence of CKD across different regions required to gather the considerable information for present and future projections are relatively sparse, while it is reported that the kidney registries in different regions of Asia have mostly been restricted to systematic data collection directly to patients with kidney failure requiring kidney replacement therapy [4]. Several individual reports from different parts of Asia, such as India, China, and Vietnam, suggest a huge variation in the prevalence of CKD [5,6].

In India, the prevalence of CKD is estimated to be 31% of people associated with diabetes [4]. CKD is accompanied by pervasive risk factors, especially in those patients having hypertension [7], diabetes mellitus [8], and depression [9], as well as acute and chronic obstructive pulmonary disease (COPD) [10]. It has been reported that an increasing projection in the number of CKD patients is not only affecting human health, but is even causing serious morbidity and mortality [11].

To perfect the treatment of CKD, several modern medicines, such as angiotensin-receptor blockers (ARBs), mineralocorticoid-receptor antagonists, sodium/glucose cotransporter 2 (SGLT2), beta-blockers, and anti-fibrotic, as well as anti-inflammatory drugs, have been utilized. Some vitamins, such as Vitamin B, C, D, etc., and amino acid-comprised drugs such as keto-analogs, are well-reported drugs for the protection of kidney function. The underlying mechanism of such drugs is reported to have a multi-mechanistic or physiological approach through inhibiting the renin-angiotensin system; normalizing glomerular hyperfiltration and low-density lipoprotein cholesterol (LDLc); reducing blood pressure, albuminuria, and systemic vascular resistance; and antioxidant defense system [12,13]. Notwithstanding the significant evolution of these therapies, the outcomes for the treatment of or even for curing kidney function are still far from perfect. Several factors such as the availability, affordability, and accessibility of drugs increase the reluctance of patients to use them, leading to their underutilization in developing or less economic countries. The majority of adverse and side effects such as gynecomastia, high potassium, fatigue, nausea, erectile dysfunction, constipation, etc., are some of the most important factors that have prompted research for the development of new medicines from nephroprotective from natural sources, which play an efficient role in the protection of the distorted function and kidney architecture [2,14].

L-carnitine (LC) is one of the essential proteins used as a supplement by recreationally active, competitive, and highly trained athletes. People with low levels of LC production in their bodies use LC as a supplement. Some people have low amounts of LC for a variety of reasons, including medical conditions such as skeletal myopathies, drug use, and genetic diseases [15,16,17,18]. Over the course of the forecast period, the market is anticipated to gain from a rising awareness of the importance of scheduling regular checks, rising public concern over health issues, and an increase in the availability of therapies for various diseases and disorders. LC has gained popularity as a functional food and beverage ingredient due to research showing that it is crucial for the treatment and prevention of a number of diseases induced by oxidative and inflammatory damage [17,19]. Customers are more likely to favor functional foods and beverages that contain LC because of their enhanced level of energy and stamina as well as other health benefits. Consumers’ growing preference for LC supplements in North America is anticipated to fuel the market’s expansion throughout the anticipated time frame [20].

Healthcare products dominated the market in 2020, accounting for more than 34.0% of total revenue. Many healthcare products, including supplements for weight loss and increased stamina, contain L-carnitine. The market for dietary supplements is anticipated to reach USD 272.4 billion by 2028 as a result of rising obesity rates, rising consumer awareness of their own health and well-being, changing lifestyles, and increasing purchasing power. Over the projection period, the manufacturers are anticipated to benefit greatly from the dietary supplement market’s quick expansion [21]. The detail of the market size and regional coverage of LC has been summarized in Table 1.

Considering the facts and economic values, the desire for the type of the drug therapies is shifting from synthetic medicine to naturally derived medicines or nutritional health supplementation. However, it has been reported that natural products or supplementation therapies have been gaining exponential growth in their utilization over the last few decades in both developed and developing countries due to availability, accessibility, affordability, natural origin, and lesser side effects. Carnitine has been acknowledged as the essential dietary nutrient synthesized biologically and is active only in the “L” isoform that contributes to cellular energy metabolism. Several studies have reported on LC and its anti-oxidant and anti-inflammatory potential and have provided an effective regimen for the amelioration of CKD [22]. Based on the above facts, the review contributes a comprehensive report on LC as a nutritional therapeutic regimen for alleviating kidney disease or related disorders.

## 2. Methods

The studies include the ideal reported or checklist from the reported research and review articles in national and international journals. Each report was shortlisted based on the rationale of our study. No reports were added in the final study that reflected the work of our original study. The study was drafted as per PRISMA guidelines.

### 2.1. Search Strategy for the Original Study

The data were assembled through a potential wide-ranging literature search using online records of Elsevier, PubMed, Google Scholar, ACS publication, Springer, etc., from available databases between 1990 and 2022 for the screening of current and impactful evidence. The keywords used in the electronic databases are as follows: CKD/kidney disease, current epidemiology and its prevalence, biosynthesis and the sources of LC, network biology, anti-oxidant and anti-inflammatory potential of LC, etc. The reference list and citation index of each screened article were checked for their impactful utility, and only the articles that possessed potentially relevant citations were included in the study. The selection criteria of the study were constrained to the language “English” because of the language barrier, high cost of translation, and time efficiency. Furthermore, to accomplish an inclusive examination strategy for the relevant information, the history of each publication and the indexing of their respective journals were checked for impactful information. The relevant data were extracted and summarized as impactful information on LC in the alleviation of CKD, therapeutic application, and future perspectives.

### 2.2. Criteria for the Original Study

#### 2.2.1. Inclusion Criteria

The inclusion criteria for the present study were restricted based on the following parameters: review or research articles published in international journals were selected for gathering the information regarding LC and CKD; its sources of LC and its therapeutic applications were the main body of the present study, while the therapeutic application was restricted to the pathophysiology of the kidneys. The reported articles that comprehensively discussed the molecular mechanism for oxidative and inflammatory stress were included in this study to understand the typical pathophysiology of CKD in the deficiency of LC.

#### 2.2.2. Exclusion Criteria

Exclusion criteria for drafting the final study were as follows: duplicate publications, irrelevant abstract or research or review reports, articles published before 1990, unauthentic reports lacking the area or localities of the study, data of untargeted disease, non-open access journals, articles that were partially accessed or not indexed or did not provide good citations were removed from the original study.

#### 2.2.3. Study Selection

All the authors assessed the studies based on their inclusion criteria by evaluating their title or abstract and by examining the full text of the study. Only well-indexed journal articles interrelated to LC and its associated myths or complications with CKD were selected for the final study in order to reflect the new and trending scientific views. The accessibility of the original study and its search output was confined to the title and the abstracts. Those articles that were related to LC and CKD or renal disease can be found at the end of the original study. Furthermore, out of thousands of published articles, only suitable articles were critically examined to enhance the original study concerning the preclinical and clinical facts that have been evaluated on LC and its supplementation. Additionally, articles related to the biochemical and molecular aspects of LC and its role against oxidative and inflammatory stress-induced CKD were added to the draft of the original study.

## 3. Review Findings

### 3.1. Sources and Biosynthesis of Creatine

Creatine is endogenous and is produced by the body at an amount of approximately 1 g/d. The biosynthesis of creatine is predominately arising in the liver and kidneys, as well as, to a lesser extent, in the pancreas. It has been reported that approximately 1 g/d of creatine is obtained from an omnivorous diet. The skeletal muscle stores approximately 95% of the body’s creatine, and a residual amount of up to 5% is disseminated in different parts of the body, such as the liver, brain, and kidney, as well as the testes. Diets that contain a predominant amount of creatine are largely meat-based (especially red meat); vegetarians have lower resting creatine concentrations [23]. LC is acknowledged as an amino acid of the non-protein category, also known as β-hydroxy-γ-trimethyl-aminobutyric acid. It is synthesized via catabolic activation of two different essential amino acids: lysine and methionine. It eases the process of the β-oxidation of different long-chain fatty acids and thus plays an important role in the metabolism of different branched-chain amino acids and steadies the cellular function [24].

Arginine is known as the starting material for the biosynthesis of creatine. The whole process occurs in the kidney and liver. During this process, the guanidino group is converted to glycine through the catalytic process by an enzyme, glycine amidinotransferase, which further processes it to produce guanidinoacetate and ornithine. Arginine and glycine aminotransferase processing is essentially started in the tubular part of the kidney, pancreas, and liver and other organs. Guanidinoacetate (GAA) is generally formed by the components of the kidney, which are methylated by guanidinoacetate N-methyltransfer in the kidneys and liver, thus producing creatine. Moreover, in the cellular compartments, creatine is converted into the proactive form of phosphocreatine. Through the conversion of the phosphocreatine, several molecules of adenosine triphosphate (ATP) are generated, which boosts the energy level by several fold [25].

Creatine synthesis is principally regulated by variations in the expression of the renal enzyme that is arginine, glycine aminotransferase, and accessibility of the substrates. Intakes of dietary creatine efficiently influence the level of circulating growth hormone (GH), which expresses the new creatine synthesis. It has been reported that guanidinoacetate N-methyltransferase is not affected by the intake of creatine supplements, or by GH, and remains the hepatic activity in function [26,27]. Therefore, the intake of creatine supplements helps to store some essential amino acids such as glycine, arginine, and methionine for further metabolic activity, which includes protein synthesis, glutathione, and NO, and this is the only reason that creatine is acknowledged for its significant nutritional and physiological importance [23]. The systematic representation of the biosynthesis of creatine has been summarized in Figure 1.

### 3.2. Perception of LC Supplementation for Kidney Dysfunction

Questions and concerns regarding the supplementation of creatine that then causes kidney dysfunction are common. It has been established that protein or LC supplementation or maximum protein intake cause renal damage. Several types of research have demonstrated no adverse effects of LC supplementation under its recommended dosages on kidney health.

However, several case studies conducted on individuals with focal segmental glomerulosclerosis reported a correlation between LC supplementation and renal dysfunctions. The outcomes of those studies confirmed that LC or any of its derived supplements do not have a negative impact on renal function. Despite the belief that LC significantly increases creatinine levels, several studies have been published that explore certain facts, even those related to it attenuating the mortality of renal tissues by oxidative and inflammatory stress [17,19,27,28,29,30,31,32,33]. In a study reported by Silva et al., LC supplementation did not significantly alter serum creatinine levels or plasma urea, and the authors concluded that LC supplementation did not promotes renal damage in the recommended amounts and durations. However, the concept that LC or its derived supplements cause kidney damage and/or renal dysfunction was not taken into consideration [34]. Moreover, a study reported by Ahmad et al. evaluated the role of LC in the amelioration or preservation of the normalcy of cognitive and renal functions in a rat model of CKD [35].

### 3.3. Network Biology and Polypharmacology of LC

LC targets the expression of several genes throughout its production and biodegradations, and it also regulates the expression of several genes such as ACADM, ACADS, CPT1A, CPT1B, CPT1C, SLC25A20, CRAT, and BBOX1, and thus medium-chain acyl-CoA dehydrogenase is an enzyme made possible by the ACADM gene (MCAD). The mitochondria, the parts of the cell that produce energy, are where this enzyme works [36]. CRAT carnitine O-acetyltransferase plays an essential role in the regulation of creatinine concentration in the blood and regulates its excretion via urine. Carnitine O-acetyltransferase, a vital enzyme in a crucial metabolic pathway that is crucial for energy balance and fat metabolism, is encoded by this gene. It is a member of the family of carnitine acyltransferases. This enzyme controls the ratio of acyl-CoA/CoA and catalyzes the reversible transfer of acyl groups from an acyl-CoA thioester to carnitine. Both the mitochondria and the peroxisome contain it [37]. Transcript variants produced by alternative splicing encode various isoforms that may localize to various subcellular compartments. TMLHE, Trimethylglycine dioxygenase, the first enzyme in the route that produces carnitine, is a protein that is encoded by this gene. The movement of activated fatty acids through the inner mitochondrial membrane depends critically on carnitine. Trimethyllysine is changed into hydroxytrimethyllysine by the encoding protein [38,39,40]. From the datamining sources through network biology conducted by Cytoscape, it was found that ACADM, CROT, CRAT, and ACADS are the most prominent genes involved in the function of LC and in the pathology of kidney disease. Furthermore, it has been described that proteins such as ACADM, ACADS, and CRAT regulate the energy production by LC. Each mined data source has been depicted in the form of a graphical representation. From the analysis of several genes, it can be demonstrated that ACADM, ACADS, CRAT, TMLHE, and CROT are the most prominent genes and have an immense role in biosynthesis and biodegradation and also provide energy kidney tissue and maintain the function of the kidney. Several enzymes, such as acetylcarnitine, malonyl-CoA, palmitoyl-CoA, and propionyl-CoA, regulate the function bioactivity of LC [15,16,41,42]. A graphical representation of the network biology involved in LC function has been depicted in Figure 2.

### 3.4. Oxidative Stress-Induced CKD

In living organism cells, free radicals in the form of ROS or RNS are normally generated due to normal cellular metabolism as well as cellular homeostasis. The excess construction of free radicals is constrained to several deleterious effects in the form of oxidative modifications and results in the dysfunction of vital cellular organelles such as oxidative autophagy, mitochondrial dysfunction, DNA impairment, etc. [43]. The excessive onset of the free and denatured proteins in the form of advanced oxidation protein products (AOPPs) counter the kidney function or the renal cortex, which is proportionated to glomerulosclerosis, podocyte injury, and tubulointerstitial fibrosis [44,45].

In oxidative stress, the cell or tissue’s ability to better resist stress impairment by preceding exposure to a lesser amount of stress is generally acknowledged as the adaptive response. However, it has been seen in all organisms against the effect of different cytotoxic agents [46]. Within the diversity of the exogenous or endogenous cytotoxic agents, one of the most abundant agents, namely oxidative stress, is known to induce an adaptive response. In this process, two mechanism approaches work together simultaneously; in the first step, in stress-regulating proteins of MAP kinase, namely MAPK10 and MAPK14A, expression is triggered. The process in turn activates other transcription factors (TFs) such as Nuclear factor NF-kappa-B (NFKB1), Nuclear factor erythroid 2-related factor 2 (NRF2), and Transcription factor Sp1, also known as specificity protein 1 (SPI). The process is simultaneously triggered by immediate early gene induction, such as JUNB and FOS transcription factors. Meanwhile, the activation of TFs induces the expression of several antioxidant enzymes that neutralize the efficacy of free radicals and maintain the normalcy of the cellular function. In the second step, the adaptive response represses the activity of those genes or the TFs that are responsible for the production of ROS. Major genes such as cytochrome P450 monooxygenase (CYP1A), Cytochrome b-245 light chain (CYBA), Xanthine dehydrogenase (XDH), and Amine oxidase (flavin-containing) A (MAOA) play an important role in the elevation of ROS or oxidative stress by inducing the expression of NOX3 to form a functional NADPH oxidase, which generates superoxide and other oxidants that are generally repressed by the adaptive response [47]. The systematic representation of oxidative stress and the role of adaptive response has been summarized in Figure 3.

Moreover, certain varieties of free radicals generated during the stressed cellular function are generally responsible for activation of the adapter protein that regulates the signal transduction initiated via the TNF receptor, along with several IL receptors followed by the activation of TNF, TTRAP, and TRAFs [4,48]. These are responsible for the degradation of cytochrome C (CYSC) oxidase, NADH dehydrogenase, and ATP production and result in shutting down the ability of mitochondria to produce energy. During this process, several initiator genes, such as BAD, APAF1, and CASP9, for apoptosis are overexpressed and activate the CASP3 as a dominant gene influencing the apoptosis of cell [49].

Activation of CYSC and CYCT leads to the expression of apoptotic and anti-apoptotic genes such as APAF1 and CASP9, which in turn activate CASP3. Activation of such apoptotic genes leads to the activation of apoptosis via the formation of a diversity of apoptotic bodies. The whole process occurs via the classical pathway for complement activation, which is comprised of a variety of complement component bodies such as C2, C4, C3a, C4a, C5a, etc. that regulate the apoptotic pathway via regulating the membrane lysis complexes that cause cell lysis and clearance [50]. During the process, the complement component activates the expression of C3a anaphylatoxin chemotactic receptor (C3AR1) and stimulates chemotaxis and granule enzyme release (GER), as well as superoxide anion production (SAP). To defend the deleterious effect of the stimulated chemotaxis, GER and SAP adaptive response induces the expression of anti-oxidant genes and stimulates the production of anti-oxidant enzymes that quench the free radicals and suppress the deleterious effects of the free radicals, thus inhibiting cell or tissue lysis. Simultaneously, the adaptive response of the body’s immune system works together via the production of enzymes or some regulatory genes such as CDKN1A, CDKN1B, and CDKN1C to normalize DNA replication, thus maintaining the ability of liver regeneration against the deleterious effect of certain varieties of free radicals of ROS (Figure 4).

LC has been acknowledged as a potential antioxidant agent; it has been reported that LC significantly scavenges free radicals, which were evaluated by several in vitro chemical-based assays [20]. At different concentrations of LC, such as 15, 30, and 45 µg/mL, 94.6%, 95.4%, and 97.1% exhibit lipid peroxidation inhibition of linoleic acid emulsion, respectively. Hence, it was proved that L-carnitine had an effect on superoxide anion radical scavenging, DPPH scavenging, hydrogen peroxide scavenging, metal chelating on ferrous ions, and total reducing power activities. The activity was comparable to the antioxidant effect of trolox and alpha-tocopherol, which were used as the reference antioxidants [31]. Cao et al. reported that LC ameliorates the efficacy of superoxide dismutase (SOD), catalase (CAT), glutathione peroxidase, and total antioxidative ability [19]. It has been reported that LC improves renal function via amelioration of oxidative damages and the anti-oxidant defensive system, and it was further suggested that L-carnitine supplementation or nutritional therapy may have some benefit in patients suffering from chronic renal failure [17].

### 3.5. Inflammatory Stress-Induced CKD

During the inflammation process, germline-encoded pattern recognition receptors (PRRs) work as the main host devices that identify the varieties of pathogen-associated molecular patterns (PAMPs) and damage-associated molecular patterns (DAMPs), and PPR activation generally triggers the stimulation of toll-like receptors (TLRs) as well as intracellular nod-like receptors (NLRs). Tumor necrosis factor-α (TNF-α) interleukin-1β (IL-1β), as well as other IL-6, are stimulated and overexpressed depending on the majority of DAMPs and PAMPs [51,52]. The process is simultaneously coordinated to obtain activation of the adopter protein, which has been acknowledged as myeloid differentiation factor-88 (MyD88). The process activates the pathway of intracellular signaling, namely nuclear factor kappa-B (NF-κB), Janus kinase (JAK)-signal transducer, and the activator of transcription (STAT) pathway, as well as mitogen-activated protein kinase (MAPK) [4,53].

During cellular stress, the expression of several proto-oncogene tyrosine-protein kinase SRC and Serine/threonine-protein kinase B-raf such as SRC, RAS, BRAF, and MEK takes place and activates the expression of extracellular-signal-regulated kinase ERKs (ERK1/ERK2). These proteins are acknowledged as binary molecular adjustments that switch the intracellular signaling pathways [54]. It is reported that RAS regulates actin cytoskeletal integrity, cell proliferation, adhesion, differentiation, and apoptosis, as well as cell migration. Furthermore, ribosomal protein S6 kinase alpha-3 (RSKs) down-regulates the expression of ERK signaling and mediates the regulation of mitogenic and stress-induced phosphorylation that causes unintentional growth, and mediates cellular proliferation, survival, and distinction by tempering mTOR signaling and suppressing the pro-apoptotic role of BAD and DAPK1 [55,56].

Transcription proteins such as p50, p52, p65, and RelA/B are coordinated by NF-κB transcription factor, and its activation is proportionated with the nature of stimuli such as PAMPs [57]. IκB kinases (IKK) is an essential part of the NF-κB transduction cascade that plays an important role in inflammation via enhancing cellular response. IκB kinase (IKK) is composed of two subunits of IκB kinase (IKKα and IKKβ and IKKγ). IκB phosphorylation regulates the expression of NF-κB [58]. The IκB protein phosphorylation activates the subsequent translocation of NF-κB to the nuclear region and arouses gene transcription. The whole process generates several kinds of inflammatory cytokines that induce inflammation and cause apoptosis and necrosis [59].

MAPKs are the serine/threonine protein kinases that activate the cellular stress caused by heat shock and mitogen osmotic stress, as well as inflammatory cytokines (such as TNF-α, IL-1, and IL-6) [60]. During this progression, mitogens and many distinction signals activate ERKs, while inflammatory cytokines stimulate the expression of JNK and p38 [61]. Several varieties of kinase proteins are generated during cellular stress, such as ERK1/2, MAP Kinase, and p38 MAP Kinase, as well as c-Jun N-terminal kinases (JNK) [60]. Overexpression of such kinase proteins stimulates cell proliferation and differentiation, which have both been seen in polycystic kidney disorder [62,63].

JAKs-associated proteins, such as JAK1, JAK2, JAK3, and TYK2 phosphorylation, derive several conformational variations to form active binding sites for STATs. The phosphorylated residues are translocated to the nucleus region, which arouses the transcription of different inflammatory genes. The progressive generation and translocation of these proteins are responsible for inflammatory stress via the generation of several varieties of cytokines and result in apoptosis or necrosis [64,65,66].

Several studies have been reported that claim the potential anti-inflammatory potential of LC or LC via the regulation of oxidative stress-induced inflammation. A study reported by Nomura et al. assessed the anti-inflammatory activity of LC supplementation against the inflammation induced in endothelial cells. The outcome of the study found that 0.5 mm of LC potentially increased both intracellular and phosphocreatine (PC) by a creatine transporter and repressed the expressions of ICAM-1 as well as the E-selectin of different endothelial tissues/cells. Furthermore, it was reported that LC downregulates the expression of adenosine A2A receptor antagonists. Hence, it can be demonstrated that LC, or its derived supplements, exhibits an anti-inflammatory effect in the contradiction of endothelial cells [32]. In a conducted-on-rat model of lung ischemia/reperfusion, five days of treatment with LC supplementation significantly decreased acute lung injury via attenuating the overexpression of TLR-4 and NF-κB. Thus, it was reported that creatine supplementation can be an effective therapeutic regimen for lowering inflammation of the lung [33]. Riesberg et al. evaluated the significant reduction of TNF-α levels by creatine against cells treated with LPS alone. During the analysis, it was also found that creatine exhibits a significant reduction in the levels of NF-κB. As the NF-κB expression is majorly dependent on the progressive expression of the TNF-α, so the nuclear extracts were examined for NF-κB pathway activation, and it was found that creatine significantly reduces the expression of the NF-κB signaling pathway and alters anti-inflammatory responses [27]. A systematic probable molecular mechanistic representation of the anti-inflammatory potential of creatine has been represented in Figure 5.

### 3.6. Safety Concerns of LC

Despites the tremendous health benefits of LC supplementations, safety assessment is one of the concerning issues for healthcare professionals or consumers. The National Institute of Health and the Office of Dietary Supplements published a report on 29 March 2021 and reported that supplemental carnitine can induce cramping in the abdomen, nausea, vomiting, diarrhea, and a “fishy” body odor at dosages of approximately 3 g per day. Rarer adverse effects include seizures in those with seizure disorders and muscular weakness in uremic individuals. According to certain studies, gut bacteria break down carnitine to create a compound called Trimethylamine N-oxide (TMAO), which might raise the risk of cardiovascular disease. People who eat meat seem to be more affected by this than vegans or vegetarians (https://ods.od.nih.gov/factsheets/Carnitine-HealthProfessional/ (accessed on 3 February 2023)).

According to the OSL risk assessment approach, intakes up to 2000 mg/day of LC equivalents for chronic supplementation show high evidence of safety, and this quantity is defined as the observed safe level (OSL). Despite the fact that many higher doses have been tried without any negative effects and may be safe, the data for ingestion beyond 2000 mg/day are insufficient to make a confident judgement of long-term safety [67,68]. Furthermore, in a sub-chronic toxicity test for 90-days on rats, it was reported that up to 50,000 ppm LC administered diets do not shows any treatment-related changes in mortality, gross pathology, hematology, ophthalmology, or histopathology. In both the presence and absence of metabolic activation, L-carnitine did not exhibit any carcinogenic activity in different bacterial strains at doses up to 5000 g/plate, and it also did not cause chromosomal abnormalities in human cells. These trials’ findings confirm that using L-carnitine and L-tartrate as a dietary source of L-carnitine is safe [68].

## 4. Discussion

In order to accomplish the loss of several kidney functions, a patient associated with CKD needs dietary adjustments that include limits of protein intake, sodium, potassium, phosphorus, etc., for the smooth function of the kidney [23]. Several foods or plant-based dietary supplements manage to evade such problems; unfortunately, such foods are unable to approach the optimal requirements of macro- or micronutrients. Normal dietary endorsements cannot avert the incidence of many symptoms, which typically include tiredness, impaired cognition, weakness, and muscle myalgia, as well as muscle wasting [69].

During treatment, one of the most concerning threats for patients associated with CKD is severe loss of proteins; low-protein diets for non-dialysis-dependent CKD and high-protein diets for dialysis-dependent CKD are often recommended to patients to evade the chronicity of protein loss [70]. Creatine deficiency is also seen in patients associated with CKD, and creatine has been acknowledged as an important supplier in the homeostasis of cellular energy. However, for a daily basis diet (average omnivorous diet), 1.6–1.7% cannot fully reimburse these losses. The endogenous production of creatine contributes an essential role in replenishing protein requirements and provides the smooth function of several vital organs. The endogenous production of creatine and enzymes such as arginine: glycine amidinotransferase (AGAT) is responsible for its production, which is facilitated in the kidney [71,72]. Progressive distortion of the renal function reduces the production of creatine and leads to its deficiency. LC supplementation can support replenishing the optimal requirements of creatine to the body for normal cellular homeostasis. However, creatine improves the smooth function of the kidney via the amelioration of renal architecture against oxidative and inflammatory stress. The activation of innate and adaptive immunity involves CD80. It has also been associated with the etiology of minimal change disorder (MCD) and a potential function for B- and T-cell mediated immunity. Urinary CD80/creatinine values were highest in MCD compared to other glomerular diseases and were increased in DN with proteinuria >2 compared to controls [73]. However, several comorbidities of the kidney can be associated with other factors, such as kidney stones or urolithiasis, etc. It has been reported that urinary extracellular vehicles (EVs) and proteins are the most prominent biomarkers that can be assessed to demonstrate hyperoxaluria [74].

Several studies have been reported that emphasize the oxidative and inflammatory stress progressively associated with sustained focal segmental glomerulosclerosis. It has been reported that L-carnitine is more efficient in decreasing inflammation and enhancing health at dosages > 2 g daily. This vitamin fights oxidative stress by lowering lipid oxidation, boosting the anti-oxidative stress defense system, and chelating metal ions [75]. A report published by Gautam et al. acknowledges the well-understood mechanism of oxidative and inflammatory stress-induced CKD [4,76]. Moreover, from the present findings, it was found that creatine or LC exhibits potent anti-oxidant and anti-inflammatory activity via regulation of several varieties of free radicals and cytokines by improving the effect of anti-oxidant enzymes such as CAT, SOD, GSH, etc., and inflammatory cytokines such as TNF, ILs, NF-Kb, etc. However, several myths are based on the recommendation of protein supplements to patients associated with CKD or renal dysfunction, as the protein supplements increase the protein and creatinine levels more than normal, effecting the glomerular filtration rate [69]. In one study, the health perspective use of carnitine supplements has been well-defined for people who are associated with CKD and require dialysis. The study revealed that the use of carnitine supplementation to treat carnitine deficit associated with dialysis is not currently supported by the data. Carnitine supplementation does not exert adverse outcomes, despite the possibility that it may usually improve anemia-related disorders [22]. Shahraki et al. studied the impact of carnitine on the nutritional parameters in patients with chronic kidney failure; however, some of the findings are still up for debate. To assess how these primary outcome affect patients’ clinical states, it is important to conduct clinical studies with better designs. With this supporting data, the potential contribution of carnitine to ameliorating the effects of malnutrition in CKD patients would be made obvious [77]. However, supplementing L-carnitine helps in the hemodialysis (HD) of dialysis patients via obviating comorbidities such as decreased exercise capacity, muscle symptoms, and increased intradialytic hypotension, as well as cardiac complications (arrhythmias, reduced output, low cardiothoracic ratio). Erythropoietin-resistant anemia was also found to ameliorate patients under dialysis, although routine administration of L-carnitine to all dialysis patients is not advised; however, a therapeutic trial of L-carnitine or Levocarnitine therapy may be helpful in symptomatic individuals with certain clinical characteristics who do not respond to standard treatments. Intradialytic muscular cramps, asthenia, hypotension, decreased oxygen consumption, cardiomyopathy, decreased ejection fraction, myopathy or muscle weakness, and anemia needing high doses of EPO are a few of these symptoms [77,78].

In a rat model of chronic CKD, a study examined the effect of l-carnitine therapy on renal function and cognitive ability. We assessed the renal function and cognitive capacities in a CKD rat model to evaluate the impact of L-carnitine on the CKD state. We discovered that all CKD animals had worsened renal function, as shown by higher blood creatinine and BUN levels, as well as several histological abnormalities. Treatment with L-carnitine dramatically lowered blood creatinine and BUN levels in CKD rats, slowed renal hypertrophy, and lessened renal tissue damage. Additionally, CKD rats displayed cognitive impairment in the two-way shuttle avoidance learning, which was reversed by the injection of L-carnitine. We found that treatment with L-carnitine dramatically enhanced cognitive and renal functioning in a rat model of CKD [35]. Levocarnitine therapy is also said to be helpful for patients under dialysis to obviate renal anemia, dyslipidemia, heart dysfunction, and muscular and dialysis symptoms [77].

A recent study reported by Salama et al. examined the L-carnitine effect against nephrotoxicity induced by potassium dichromate (PD) in rats via modulation of the PI3K/AKT signaling pathway. In comparison to normal rats, the administration of PD led to raised levels of malondialdehyde, tumor necrosis factor-alpha, and transforming growth factor-beta in the renal tissue, as well as elevated levels of serum creatinine and blood urea nitrogen. Additionally, PD caused histological damage indicative of apoptosis and downregulated of the PI3K/Akt signaling pathway, indicating continuing apoptosis. All of the aforementioned blood, renal tissue, and histological parameters in the current study were improved when L-carnitine was used at the chosen dose levels in comparison with nephrotoxic rats. After using PD, the PI3K/Akt signaling pathway was downregulated, whereas L-carnitine increased it. Hence, it was concluded that through activation of the PI3K/Akt signaling pathway, L-carnitine greatly reduced the risk of apoptosis, signaled the return of normal renal cell growth and function [78].

In several case studies, it was found that no significant changes in the level of creatinine have been found in those patients who were associated with CKD [18]. However, the typical symptoms in a CKD patient, such as tiredness, impaired cognition, muscle weakness, myalgia, and muscle wasting, were found significantly attenuated and improved the health of patients as well as the kidney [69]. Furthermore, the recommended dose of creatine based on the severity of renal dysfunction or CKD played an essential role in improving patient health. Therefore, it can be suggested that LC supplementation in a CKD patient with the recommended dose helps to manage the renal function, not only via replenishing the nutritional requirement but also by improving the anti-oxidative and anti-inflammatory defensive system. Furthermore, from the previous findings it can be suggested that the most beneficial and recommended dose of LC is demonstrated as 2000 mg/kg. LC supplementation is typically tolerated because it is quickly excreted from the body. Pre-clinical and clinical investigations have shown the most beneficial effects of LC when it is employed as medication [67,68].

## 5. Conclusions

The emerging evidence suggests that the severity of focal segmental glomerulosclerosis is majorly associated with oxidative and inflammatory stress and extensive fluctuations in biochemical markers. Hemodialysis patients are generally affected by decreased exercise capacity, muscle symptoms, and increased intradialytic hypotension, as well as cardiac complications (arrhythmias, reduced output, low cardiothoracic ratio). Erythropoietin-resistant anemia is a major occurrence in hemodialysis patients. From extensive research and analysis, it can be demonstrated that LC or creatine supplementation provides an effective adjuvant or therapeutic regimen that significantly reduces oxidative and inflammatory stress and erythropoietin-resistant anemia, and it also evades comorbidities such as tiredness, impaired cognition, muscle weakness, myalgia, and muscle wasting. However, no significant changes were found in biochemical alteration, such as creatinine, uric acid, urea, etc., after creatine supplementation in patients with renal dysfunction. The expert-recommended dose of LC or creatine to a patient is approached for better outcomes of LC as a nutritional therapy regimen for CKD-associated complications.

Despite much study having been done to explore the most beneficial prophylactic dosages of carnitine in a number of disease scenarios, there is significant controversy and misunderstanding regarding its role in normal nutrition. The most beneficial and recommended dose of LC is demonstrated as 2000 mg/kg. LC supplementation is typically tolerated because it is quickly excreted from the body. Pre-clinical and clinical investigations shows the most beneficial effects of LC when it is employed as medication.

## Figures and Tables

**Figure 1 jpm-13-00298-f001:**
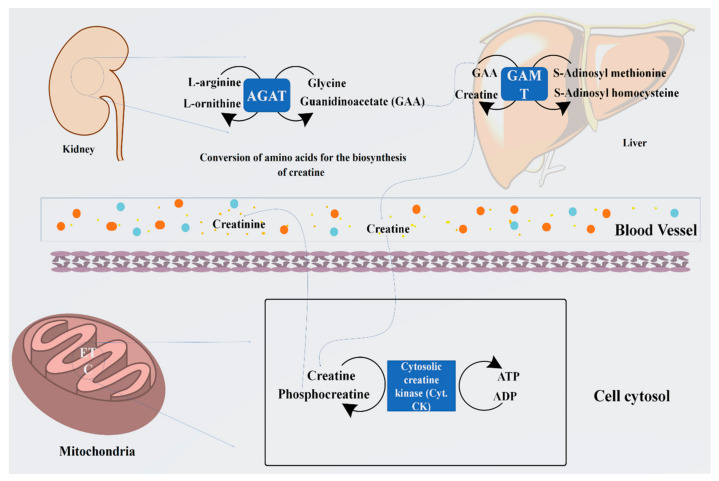
Representation of biosynthesis of creatine from arginine and glycine by the enzyme arginine: glycine amidinotransferase (AGAT) to guanidinoacetate (GAA) and its methylation by guanidinoacetate N-methyltransferase (GAMT) using S-adenosyl methionine to form creatine. Furthermore, its cellular compartment conversion into phosphocreatine (PCr) by the cytosolic enzyme creatine kinase produces several folds of energy, such as ATP.

**Figure 2 jpm-13-00298-f002:**
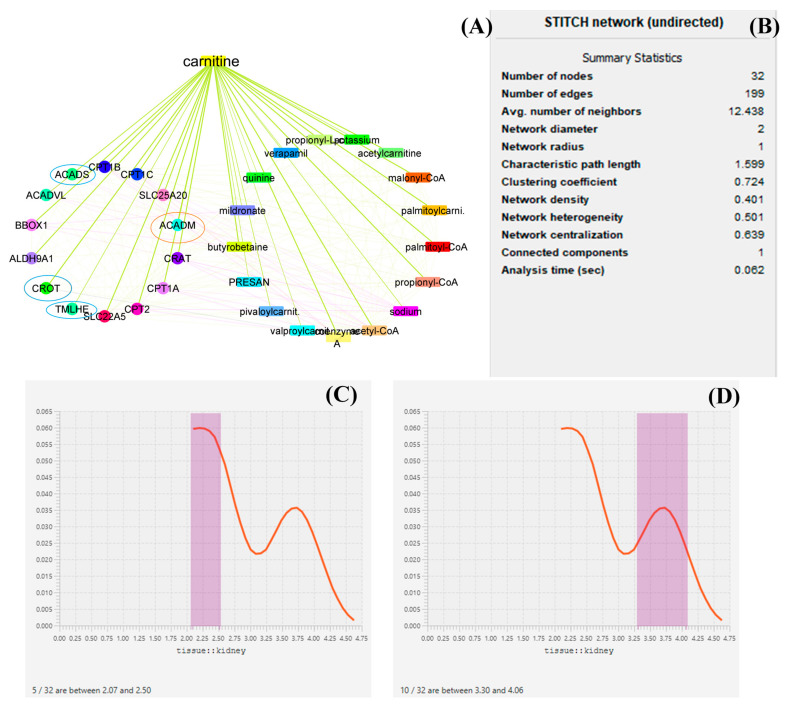
Network biology and pharmacology of LC based on datamining sources. the study revealed the biomolecular role of LC based on its network as well as gene pathology. in this diagrammatic representation, (**A**) represents the biological networks and their interaction with the genes involved in LC biosynthesis, function, the provision of energy, and biodegradation. (**B**) represents the statistical summary of CPI network, this network possesses 32 nodes with 199 interacted edges. (**C**,**D**) represent a histogram of the most prominent genes and their roles in maintaining kidney function. In (**A**), the red circle represents histogram (**D**) and the blue circle represents histogram (**C**). The genes under the circle are characterized as the most prominent genes involved in pathophysiology.

**Figure 3 jpm-13-00298-f003:**
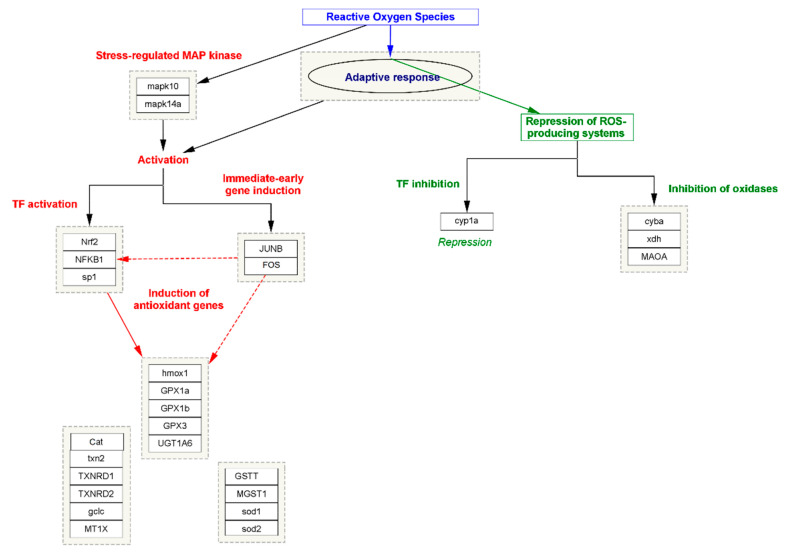
The systematic representation of the biomolecular approaches involved in oxidative stress and the role of adaptive response.

**Figure 4 jpm-13-00298-f004:**
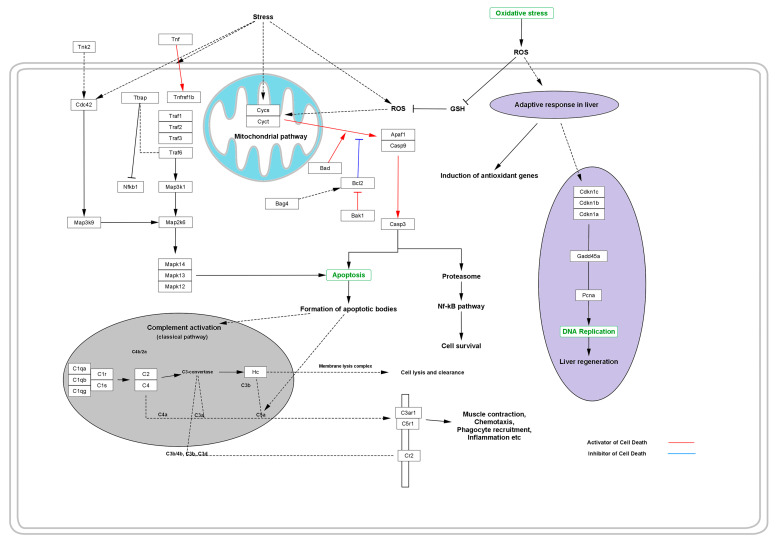
Effect of oxidative stress in the progressive distortion of the anti-oxidative system and adaptive response, thus replenishing the cellular protection from programmed cell death or necrosis.

**Figure 5 jpm-13-00298-f005:**
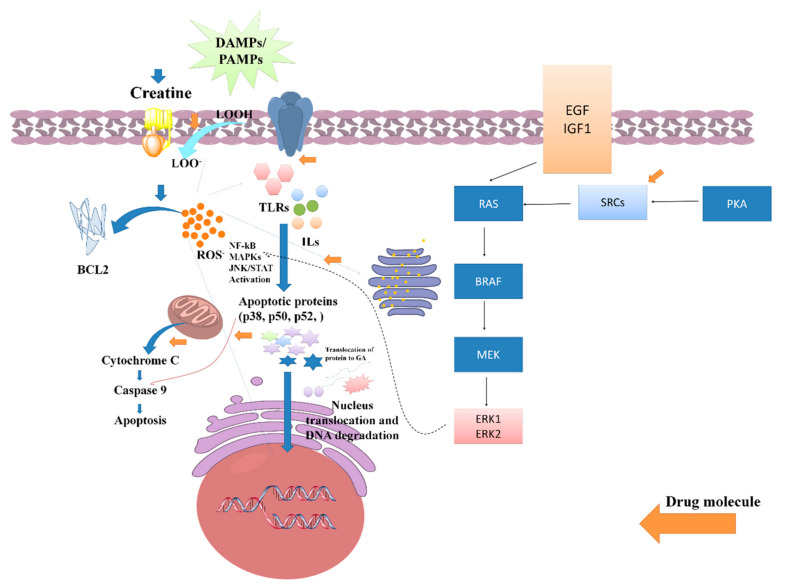
Role of creatine against PAMPs and DAMPs induced inflammatory stress and its impact on various signaling pathway activations such as NF-kB, MAPKs, and JAN/STATE.

**Table 1 jpm-13-00298-t001:** Scope of L-carnitine, market research, and regional coverage.

Report Attribute	Details
Base year for estimation	2020
Country scope	USA; Canada; Mexico; Germany; UK; France; Italy; China; India; Japan; South Korea; Brazil; Argentina; South Africa; Saudi Arabia
Customization scope	Free report customization (equivalent up to eight analyst working days) with purchase. Addition or alteration to country, regional, and segment scope.
Forecast period	2021–2028
Growth Rate	CAGR of 5.1% from 2021 to 2028
Historical data	2017–2019
Key companies profiled	Lonza (Basel, Switzerland); Northeast Pharmaceutical Group Co., Ltd. (NEPG) (Shenyang, China); Biosint S.p.A. (Sermoneta, Italy); Cayman Chemical (Ann Arbor, MI, USA); Merck KGaA (Darmstadt, Germany); Tokyo Chemical Industry Co., Ltd. (Tokyo, Japan); Ceva (Marseille, France); Kaiyuan Hengtai Nutrition Co., Ltd. (Kaiyuan, China); ChengDa PharmaCeuticals Co., Ltd. (Jiaxing, China); Huanggang Huayang Pharmaceutical Co., Ltd. (Huanggang, China); HuBeiYuancheng SaichuangTechnology Co., Ltd. (Wuhan, China)
Market size value in 2021	USD 194.0 million
Pricing and purchase options	Avail customized purchase options to meet exact research needs. Explore purchase options
Quantitative units	Volume in tons, revenue in USD thousand, CAGR from 2021 to 2028
Regional scope	North America; Europe; Asia Pacific; Central and South America; Middle East and Africa
Report coverage	Volume forecast, revenue forecast, company ranking, competitive landscape, growth factors, and trends
Revenue forecast in 2028	USD 276.0 million
Segments covered	Process, product, application, region

## Data Availability

Not applicable.

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
