# Peer review of "L-Carnitine and Chronic Kidney Disease: A Comprehensive Review on Nutrition and Health Perspectives"

_jpm, 2023, doi:10.3390/jpm13020298_

Round 1

Reviewer 1 Report

Review of jpm-2091814 

L-Carnitine and Chronic Kidney Disease; A comprehensive review on its nutrition and health Perspective by Bharti Sharma , Dinesh Kumar Yadav 

 This review aims to understand L-carnitine as an adjunctive therapy for CKD and its complications from a health perspective.

However, the title, contents, and conclusion are not consistent with the discussion. I think it would be better not to include L-carnitine in the title.

There is an argument that protein intake is important for CKD, but this is very poorly supported by this content. Even now, there are ongoing debates in CKD regarding the amount of protein intake, and the conclusions of those debates cannot be determined by your text.

If you are going to recommend creatine or L-creatine supplementation for CKD patients, you need to detail the results of your reviewed study.

・There is no explanation or A, B, C, and D in Figure 2.

Author Response

Reviewer 1

Comments and response

Comments

Response

This review aims to understand L-carnitine as an adjunctive therapy for CKD and its complications from a health perspective.

However, the title, contents, and conclusion are not consistent with the discussion. I think it would be better not to include L-carnitine in the title.

There is an argument that protein intake is important for CKD, but this is very poorly supported by this content. Even now, there are ongoing debates in CKD regarding the amount of protein intake, and the conclusions of those debates cannot be determined by your text.

If you are going to recommend creatine or L-creatine supplementation for CKD patients, you need to detail the results of your reviewed study.

Dear Sir/Madam,

Thank you for your valuable suggestions.

In this study, we have significantly contributed to add the scientific aspects of L-carnitine in relation to CKD. However, the possible findings have been further added to enhance the credibility and understanding of the readers as per suggestions.

There is no explanation or A, B, C, and D in Figure 2.

The caption of figure 2 has been revised as per suggestions.

Reviewer 2 Report

In this study, the authors tried to comprehend the health perspective of L-Carnitine (LC) as adjuvant regimen for alleviating CKD and its associated complications. A detail explanation on biosynthesis of Carnitine, and its role in CKD has been defined along with biomolecular approached. Perception for creatine supplementation for kidney dysfunction has been also well defined and provide the scientific evidences on negative myth of Carnitine that it progresses kidney disease. This review has well explored about nutrition and kidney health Perspective of Carnitine. Although, there are some minor changes, that needs to be incorporated in the manuscript before publication to improve it further.  

·   Kindly re-write the sentence “The studies include the ideal reported or checklist from the reported research and review articles in National and International journals. Each report was shorted based on the rationale of our study. No reports were added in the final study which were reflecting the work out of our original study. The study was drafted as per PRISMA guideline” not giving the clear meaning

·       What are the inclusion criteria of the study?

·       How did the author do network biology or pharmacology analysis?

· I would suggest authors to cite these articles ( https://doi.org/10.1016/j.ekir.2020.08.001,

https://doi.org/10.1186/s12882-021-02417-8 and  https://doi.org/10.1186/s13023-020-01607-1  in the introduction section, to make to make it more clarity.

·       What is the significance of Figure 2A, why the authors marked with circle?

·       What do you mean by this “Carnitine O-acetyltransferase, a vital enzyme in a crucial metabolic pathway that is crucial for energy balance and fat metabolism, is encoded by this gene” Kindly re-write this.

· Kindly maintain the uniformity of the abbreviations throughout the manuscript. Eg: TNF, ILs, NF-Kb.

·   Kindly elaborate the conclusion section and correlate the role of Carnitine with respect to CKD.

Overall, Its very informative.  

Author Response

Reviewer 2

Comments and response

Comments

Response

Kindly re-write the sentence “The studies include the ideal reported or checklist from the reported research and review articles in National and International journals. Each report was shorted based on the rationale of our study. No reports were added in the final study which were reflecting the work out of our original study. The study was drafted as per PRISMA guideline” not giving the clear meaning

Dear Sir/Madam,

Thank you so much for your suggestions in context of our manuscript.

The paragraph has been changed as per suggestion to make it more understandable with clear mean.

What are the inclusion criteria of the study?

The articles published at national and international scientific journals related to Carnitine and chronic kidney disease were searched and the impactful information was added.

How did the author do network biology or pharmacology analysis

Network pharmacology study was performed using Cytoscape software 2.8.3. version.

I would suggest authors to cite these articles ( https://doi.org/10.1016/j.ekir.2020.08.001,

https://doi.org/10.1186/s12882-021-02417-8 and  https://doi.org/10.1186/s13023-020-01607-1  in the introduction section, to make to make it more clarity.

The suggested citations have been incorporated in the text as per suggestions.

What is the significance of Figure 2A, why the authors marked with circle?

There is typo error in figure serial number. The discussed figure is represented as Figure 3. And in this figure, Figure 3A, the circle represents the potential genes that are involved in regulation of the pathophysiology of kidney.

What do you mean by this “Carnitine O-acetyltransferase, a vital enzyme in a crucial metabolic pathway that is crucial for energy balance and fat metabolism, is encoded by this gene” Kindly re-write this.

The sentence has been revised as per suggestion.

Kindly maintain the uniformity of the abbreviations throughout the manuscript. Eg: TNF, ILs, NF-Kb.

The uniformity of the abbreviations throughout the manuscript has been maintained as per suggestion.

Kindly elaborate the conclusion section and correlate the role of Carnitine with respect to CKD.

The conclusion section has been revised as per suggestion. The changes has been marked in the revised manuscript.

Reviewer 3 Report

The review is aimed to comprehend the health perspective of L-Carnitine (LC) as adjuvant regimen for alleviating CKD and its associated complications. 

There are some minor comments that should be addressed before its publication. 

This study used a large amount of space to comprehensively discuss the molecular mechanism of oxidative and inflammatory stress to understand the typical pathophysiology of CKD in deficiency of LC. However, only a few articles on LC application in CKD were included, and only the conclusions were described. It is suggested to describe the specific research methods, population and results to enhance the strength of the evidence. I suggest an extensive revision with more explanation of the results of LC in CKD.

Author Response

Reviewer 3

Comments and response

Comments

Response

The review is aimed to comprehend the health perspective of L-Carnitine (LC) as adjuvant regimen for alleviating CKD and its associated complications.

There are some minor comments that should be addressed before its publication.

This study used a large amount of space to comprehensively discuss the molecular mechanism of oxidative and inflammatory stress to understand the typical pathophysiology of CKD in deficiency of LC. However, only a few articles on LC application in CKD were included, and only the conclusions were described. It is suggested to describe the specific research methods, population and results to enhance the strength of the evidence. I suggest an extensive revision with more explanation of the results of LC in CKD.

Dear Sir/Madam,

Thank you so much for your valuable suggestions. We appreciate your comments on our manuscript for its betterment.

The necessary changes and the appropriate references have been cited in the revised version of the manuscript.

Round 2

Reviewer 1 Report

I appreciate that the text is well revised.

It is important to also state regarding L-carnitine overdose, which should prompt a warning.

Specific amounts that may cause adverse events should also be indicated.

Author Response

Reviewer 1

Comments and response

Comments

Response

It is important to also state regarding L-carnitine overdose, which should prompt a warning.

Specific amounts that may cause adverse events should also be indicated.

Dear Sir/Madam,

Thank you for your valuable suggestions.

The Safety concerns of LC has been added in 3.6. section of the revised manuscript. From the possible findings it was found that as per the OSL risk assessment method indicates that the evidence of safety is strong at intakes up to 2000mg/day l-carnitine equivalents for chronic supplementation, and this level is identified as the OSL. Although much higher levels have been tested without adverse effects and may be safe, the data for intakes above 2000mg/day are not sufficient for a confident conclusion of long-term safety.
